# Changes in Challenging Behaviors Accompanying Transition to a New Facility in Individuals with Intellectual Disabilities

**DOI:** 10.3390/healthcare12030288

**Published:** 2024-01-23

**Authors:** Mizuho Kawanaka, Yuto Iwanaga, Akiko Tokunaga, Toshio Higashi, Goro Tanaka, Akira Imamura, Ryoichiro Iwanaga

**Affiliations:** 1Department of Occupational Therapy Science, Nagasaki University Graduate School of Biomedical Sciences, 1-7-1 Sakamoto, Nagasaki 852-8520, Japaniwanagar@nagasaki-u.ac.jp (R.I.); 2Service Promotion Section Business Support Division, Nanko Airinkai Social Welfare Corporation, Fukudamachi, Isahaya 854-0001, Japan; 3Nagasaki City Disability Welfare Center, 2-41 Morimachi, Nagasaki 852-8104, Japan

**Keywords:** challenging behavior, residential facility, environment, intellectual disabilities, group home, autism spectrum disorder

## Abstract

Challenging behavior (CB), the most common example being extreme self-injurious or aggressive/destructive behavior, is often observed as a major behavior issue in individuals with severe intellectual disabilities. This study investigated how CB changed among residents of a facility for people with disabilities before and after it was restructured from a traditional format single room shared by two to three individuals with approximately 20 residents lived together to a format featuring private areas with two rooms per resident and a unitcare system. Twenty-one residents of Care Home A, which was rebuilt in the new care format, were selected. Care staff completed a questionnaire one month before, one month after, and six months after residents moved to the new facility. Scores were compared among each time point. The results revealed significant reductions in residents’ aggressive, stereotyped, and targeted behaviors, such as hitting their own head and fecal smearing. The major features of the restructured facility were a living space consisting of two private rooms per resident and a shift to unit care for the entire ward. These new features enabled residents to reduce destructive stimuli and made it easier to understand what to do in each private room.

## 1. Introduction

Challenging behavior (CB), such as extreme self-injurious, aggressive/destructive behavior or sleep disturbances, is common among people with severe intellectual disability or autism spectrum disorder (ASD) [1]. Benson et al. reported that around 50% of people with intellectual disabilities displayed some form of CB [2]. Furthermore, CB may lead to negative impacts on health for individuals demonstrating CB and to negative emotional reactions and burnout syndrome among their caregivers [3,4]. Children with comorbid ASD and intellectual disability are more likely to experience abuse and display aggression, hyperactivity, and tantrums than children with typical development. Because CB makes it difficult for caregivers to raise children, it can be a risk factor for abuse [5]. 

However, an individual’s disability alone does not cause CB. CB results from interactions between disability characteristics and physical and social environmental factors [6]. Therefore, environmental accommodations tailored to specific disability characteristics would effectively reduce CB. Previous studies that examined how adjustments to the environment have influenced the behavior of people with intellectual disabilities and ASD include the following: A meta-analysis of studies using the TEACCH method, which is an intervention method tailored to learning styles associated with ASD, found reductions in inappropriate behaviors among children with ASD [7]. In a study of preschool children with ASD, it was found that an intervention that added the personal work system and visual clues of the TEACCH program to regular rehabilitation showed childrens’ improvement in problem behavior scores compared to regular rehabilitation [8]. Another study on an intervention using a structured method (one of the TEACCH program strategies) at residential facilities reported reduced stereotyped behaviors and inappropriate speech [9]. In addition to the use of visual schedules and communication tools, these studies also used methods of structuring the physical environment. Therefore, it becomes challenging to know the specific impacts that physical environmental adjustments alone have in reducing behavioral disorders.

Furthermore, sensory processing difficulties, which are included in the diagnostic criteria for ASD, are reportedly associated with maladaptive behaviors [10]. Environmental accommodations such as the use of earphones to block outside noise have been shown to improve maladaptive behavior among children with ASD [11]. Blocking environmental noise using noise-canceling headphones is known to lower sympathetic nervous system activation [12]. For adults with ASD, conducting sessions in a Snoezelen room (a room where one can experience a variety of sensory stimuli) helps reduce the severity of ASD, as evaluated using the Childhood Autism Rating Scale (CARS) [13]. Although support to reduce unpleasant sensory stimuli has proven effective in addressing CB, there remains a paucity of studies demonstrating its effectiveness in a more daily life context.

As previously mentioned, it was suggested that CB is influenced by environmental factors. Therefore, we need to consider the environmental aspects of individuals with CB, encompassing not only their school and activity settings but also their residential situations. Several studies have explored the relationship between living environments and resident behaviors. Bhaumik et al. (2009) reported that using a person-centered approach by a team of medical/welfare professionals to facilitate the transition to life in the community after discharge from long-term care helped reduce the incidence of aggressive behaviors after reintegration into the community [14]. Olivier-Pijpers et al. (2020) also showed that a facility’s physical environment and a low staff turnover rate influenced CB severity [15].

Moreover, when searching for studies on residential facility living format, research on older adult care reported an increased quality of life (QOL) among residents [16], the facilitation of occupational performance [17], and reduced feelings of emotional exhaustion among care staff [18] as a result of providing care in a facility comprising “units”, each with a small number of residents, rather than a traditional structure in which a large number of residents receive care all at once in a single facility. Research on people with intellectual disabilities has also reported improvements in resident QOL at personalized residential facilities compared to traditional ones [19].

These findings demonstrate that living in a residential facility that employs a small group “unitcare” format with structured spaces is beneficial for children and adults with CB, many of whom use residential facility services and display problematic behavior as a result of interaction with their environment. However, thus far, no study has examined behavioral changes over time among residents with CB who transition from a traditional, large group, communal living-style facility to one with a structured living environment that accommodates their disability characteristics and provides “unit care”. 

This study investigated the changes in residents’ CB from one month before transitioning from a conventional format where multiple people shared one room, and several residents lived together on a floor, to one month and six months after transitioning to a unitcare, apartment format, in which five residents lived together in one unit which was further separated into five private apartments comprising two divided rooms per apartment. The purpose of this study was to investigate whether the transition to a new facility that takes into account disability characteristics reduces CB and whether changes in CB persist after the transition.

## 2. Materials and Methods

### 2.1. Study Facility

#### 2.1.1. Before Restructuring

We surveyed Care Home A, which was scheduled to be rebuilt. Care Home A included two buildings: a men’s ward and a women’s ward. The men’s ward was erected in 1970, and the women’s in 1996. Both buildings were two stories and employed a traditional facility structure. In this structure, the residents’ rooms were arranged in a single line facing a hallway, with a dining hall at the end and a restroom in the middle.

Living spaces for residents included two single occupancy rooms and 22 rooms housing two to three residents. Rooms used by multiple residents did not contain partitions or separate spaces for individual residents.

The men’s ward had two restrooms on the first floor and one on the second floor, whereas the women’s ward had two restrooms on the first and second floors. Each restroom had two or three toilets. This arrangement meant that, sometimes, residents could not use the restroom when needed. 

The facility had four dining halls: two in the men’s ward and two in the women’s ward. In each dining hall, 10–15 residents gathered at a time for meals. In the men’s ward, residents who did not require special assistance with eating (approximately 10 individuals) sat in two rows at one large dining table. 

#### 2.1.2. After Restructuring

The restructured facility, completed in 2021, was erected on the same ground adjacent to the old facility, maintaining no change in the environment outside the building. The new building had two floors consisting of eight units, with five residents living in each unit. Each unit had a restroom and dining space, which shortened the distance residents needed to walk. 

Each resident had a private apartment of two rooms. The apartment was partitioned into a living room, where residents could watch TV, listen to music, or engage in other hobbies, and a bedroom. Thus, the areas for activities and rest were physically partitioned by a wall, implementing the concept of a physically structured space. The entrances to each resident’s room displayed their favorite pictures alongside their names, making it easier to distinguish individual rooms visually. 

Each unit had two private restrooms equipped with a Western-style toilet close to the residents’ private apartments. This new configuration provided residents with access to roughly one private restroom for every two residents, thereby reducing stimulation from other residents and enabling them to use the restroom freely. 

Instead of a dining hall, the five residents of each unit ate together in a single room. More seats were facing the wall, reducing unnecessary stimulation during mealtime (Table 1).

### 2.2. Participants

The participants were 37 residents of Care Home A. T consent to conduct this study was obtained from the director of Care Home A. The consent form was distributed to the legal guardians of the residents, and only those who responded affirmatively, agreeing to participate in the study, were included. We observed that these 21 who had guardian consent were found to have behavioral problems in the preliminary survey described below. All measures were completed by the care staff, who provided direct care to the residents during the observation period (8:00 a.m.–5:00 p.m.). Excluding the few staff members who were replaced after restructuring, the same staff members provided care before and after the transition, with no other changes.

### 2.3. Measures

#### 2.3.1. Preliminary Survey

Basic characteristics

Questions included residents’ sex, age, level of disability recorded in the Certificate of Intellectual Disability, and diagnosis. The Certificate of Intellectual Disability is a document used in the Japanese social welfare system based on the severity of an individual’s intellectual disability. Severity was classified as profound, severe, moderate, or mild based on IQ and adaptive behavior [20].

2.Questionnaire on problematic behaviors

The care staff described up to three resident behaviors they found difficult to deal with. Participants provided a specific description for each behavior, its frequency over the past month, and the times when the behavior was more likely to arise. Based on these responses, problematic behaviors were defined in detail, and residents displaying such behaviors at least once per week were selected as survey participants. 

3.Behavior-Related Items

Behavior-Related Items is an assessment scale developed by the Japanese Ministry of Health, Labor, and Welfare to assess the severity of CB and provide welfare services to those with particularly severe CB. It comprises 12 items inquiring about the frequency and severity of behaviors. Higher scores indicate more severe CB [21]. A study was conducted on the association between Behavior-Related Items and the Japanese version of the Aberrant Behavior Checklist (ABC-J) [22], a scale used to measure the extent and severity of CB. The study found a significant difference in behavior-related item scores when participants were classified into groups according to the extent of CB and their ABC-J scores [23]. 

#### 2.3.2. Primary Survey

In addition to the Behavior-Related Items scale, the care staff completed the Behavior Problems Inventory-Short Form (BPI-S) and a behavior observation sheet [24]. They observed residents’ behaviors and completed three questionnaires for one week, corresponding to one month before, one month after, and six months after the residents moved. Female residents moved in March 2021, and male residents moved in June 2021. The survey timeline based on these data is shown in Figure 1.

Behavior Problems Inventory-Short Form

The BPI-S is a shortened version of the BPI-01 [25], developed to quantify maladaptive behaviors expressed by people with intellectual disabilities. It comprises three scales: self-injurious behavior, aggressive/destructive behavior, and stereotyped behavior. Self-injurious and aggressive/destructive behaviors were evaluated for frequency and severity, whereas stereotyped behaviors were evaluated only for frequency. Higher scores indicated greater behavioral frequency and severity. Rojahn et al. confirmed the reliability and validity of the English version using clinical data [26]. The Japanese version created by Inoue [27] was confirmed to have adequate test–retest reliability, inter-rater reliability, and criterion-related validity.

2.Behavior Observation Sheets

The authors created a behavior log sheet for each resident based on their responses to a questionnaire on problematic behaviors in the preliminary survey. The care staff logged the frequency and location at which they observed problematic behaviors. Regarding frequency, the authors defined what one instance of behavior consisted of, and the care staff provided a count based on that definition. For example, one resident’s aggressive behavior was a challenge for the staff. Then, based on the answers of the preliminary survey, we defined this aggressive behavior as “pulling another resident or staff member so forcefully that the person being pulled cannot release their grip with normal strength, or engaging in violent behavior such as hitting”. The count was based on the number of times a person’s arm was grabbed and held until they let go or the number of times another person was struck. Each location was assigned a set code which care staff included alongside the frequency number. The observation period was the facility’s day shift from 8:00 a.m. to 5:00 p.m. The care staff observed and recorded the type and frequency of participants’ behaviors each time they completed their hourly rounds. The sheets also included a column to record the severity (the extent to which care staff wanted the behavior to improve) and difficulty (the extent to which care staff found the behavior to be difficult) of each behavior because these scores are needed for the calculation of the Goal Attainment Scale (GAS) T-score in later analyses (Figure 2). Since each resident had a different number of behaviors posing unique challenges, a T-score was calculated to provide a comprehensive rating of the achievement of multiple goals. 

The observed behaviors were converted into scores using the GAS. The GAS, developed in 1968 by Kiresuk and Sherman, is used to score the extent of goal achievement. The baseline is −1 point. If the goal achievement status declined compared with the baseline, a score of −2 points wasi assigned. If a goal is achieved, a score of 0, +1, or +2 points is assigned according to the degree of achievement [28]. The authors set the goal of reducing the frequency of residents’ problematic behaviors and defined statuses corresponding to −2 points through +2 points. The degree of goal achievement was evaluated using a T-score. Conversion to a T score was performed using the following formula: T = 50 + 10 Σ (Wi Xi)/{0.7 Σ Wi^2^ + 0.3 (Σ Wi)^2^}^1/2^ [29]. Wi represents the weight assigned to the ith goal, which is calculated as the product of the severity and difficulty scores described in the previous paragraph. Severity and difficulty were scored on a scale of 0 to 3, where 0 points indicate not at all severe/difficult, 1 point indicates a little severe/difficult, 2 points indicate moderately severe/difficult, and 3 points indicate very severe/difficult. Xi is assigned a score for the degree of achievement of the ith goal.

### 2.4. Statistical Analysis

The scores for all the scales were compared using SPSS ver. 20.0. The Wilcoxon signed-rank test was selected due to the nonparametric nature of the data. The analyses compared scores 1 month before moving to 1 month after, and 1 month before moving to 6 months after, to investigate how transitioning to a new facility influences behavior. Additionally, the comparison between 1 month after moving to 6 months after moving was included to determine whether the behavioral changes were sustained. Thus, this study involved multiple comparisons using the Wilcoxon signed-rank test. However, adjusting the alpha level was necessary because conducting multiple hypothesis tests simultaneously increases the risk of false positives. To address this, we employed the Bonferroni correction with an alpha value of 0.01667, resulting in an adjusted significance level of *p* < 0.017. Behavior observation sheets were compared using GAS T-scores. The effect size (r) was calculated for each comparison.

This study was approved by the Ethics Committee of the Health Sciences Division of the Graduate School of Biomedical Sciences, Nagasaki University (Approval no. 20121002, on 12 February 2021).

## 3. Results

### 3.1. Basic Characteristics

The selected residents were 21 out of the 37 that satisfied the eligibility criteria, comprising 17 men and 4 women with a mean age of 35.7 ± 12.1 at the time of the preliminary survey. Eleven patients received a doctor’s diagnosis of ASD, and twenty-one were diagnosed with intellectual disability. Intellectual disability severity was profound in fifteen participants, severe in four, moderate in one, and there was no answer for one. The mean length of stay was 11 years 1 month ± 13 years 2 months. 

Based on the preliminary survey, the most common problematic behavior observed was acts of harm to others, such as violence or destroying things in nine participants. As noted in seven participants, this behavior was followed by behavioral problems related to toileting, such as fecal smearing, urination, or defecating in places other than the toilet. Other behavioral problems included yelling, going outside without permission, and fixation. However, each of these behaviors was observed in four or fewer participants (Table 2).

### 3.2. Changes in Residents’ Scores on Behavior-Related Scales

#### 3.2.1. Behavior-Related Items

The total score for Behavior-Related Items decreased significantly from one month before moving to one month after (*p* = 0.007, r = 0.44). 

#### 3.2.2. BPI-S Scores

The frequency score for the self-injurious behavior scale decreased significantly from one month before moving to six months after moving (*p* = 0.031, r = 0.42). In contrast, the severity score decreased significantly from one month after moving to six months after moving (*p* = 0.030, r = 0.51). The frequency and severity scores for the aggressive/destructive behavior scale decreased significantly from one month before moving to one month after moving (*p* = 0.003, r = 0.66; *p* = 0.003, r = 0.68, respectively) and from one month before moving to six months after moving (*p* = 0.001, r = 0.70; *p* = 0.006, r = 0.69, respectively). The frequency score for the stereotyped behavior scale decreased significantly from one month before moving to one month after moving (*p* = 0.008, r = 0.58) and from one month before moving to six months after moving (*p* = 0.004, r = 0.63). 

#### 3.2.3. Behavior Observation Sheets

Based on the behavioral observation sheets, the GAS T-scores increased significantly from one month before moving to one month after moving (*p* = 0.006; Figure 3). 

## 4. Discussion

This study investigated how residents’ CB changed as a result of moving from a residential facility in a traditional format to a facility practicing unit care tailored to residents’ disability characteristics. The results demonstrated a reduced frequency of CB overall one month after transitioning. The reduction in aggressive/destructive and stereotyped behaviors persisted after six months. 

Tola et al. (2021) highlighted the following aspects to be considered when designing a facility for children or adults with ASD: (1) sensory quality, (2) intelligibility, and (3) orientation [30]. (1) Sensory quality involves addressing one of the diagnostic criteria for autism in the DSM-5, which is “excessive or low response to sensory stimuli”, and creating an environment to reduce the impact of sensory stimuli. In the new facility, a unit structure was implemented with a small number of residents, and residents were able to use private rooms and restrooms alone. These features are believed to reduce the amount of sensory stimulation from other residents, aligning with the first aspect (1). Sensory processing difficulties have been linked to reduced adaptive behaviors, which may have also played a role in this study [31].

(2) Intelligibility refers to the ease of understanding and use for individuals with ASD. In the new facility, resident rooms were designed with two separate spaces, clearly distinguishing areas for rest and hobbies. In addition, residents’ favorite illustrations were displayed at the entrances to their rooms, making it easy to differentiate between rooms. These features align with aspect (2), intelligibility.

(3) Orientation pertains to how easily individuals with ASD can voluntarily move to a desired location. The innovations in aspect (2), intelligibility, contributed to the clarity of the meaning of each space. Moreover, the transition to a unit-type structure reduced the distance users needed to travel to reach each space, falling under aspect (3), orientation. Shortening the travel distance is thought to have minimized individuals’ exposure to unwanted stimuli during travel, making it simpler for individuals to locate their desired destinations. As the new facility was designed with due consideration of these aspects, it could have helped decrease problematic behaviors one month after moving into the new facility. 

In this study, aggressive/destructive and stereotyped behaviors, in particular, decreased one month after moving, and this decrease persisted at six months. Im (2021) reviewed the treatment methods for aggressive behavior in individuals with ASD and reported that the primary methods currently supported by evidence are pharmacotherapy and therapeutic exercise [32]. This study demonstrates that environmental accommodation may also be effective in reducing aggressive behaviors among adults with ASD or intellectual disabilities. Embregts et al. (2009) analyzed the function of residents’ aggressive behavior in residential facilities and found that it often serves a social purpose. For example, getting the attention of the target of aggression or requesting something the resident wants from another person [33]. The unitcare lifestyle in the new facility likely made it easier for residents to make requests to others because of the staffing. It was, therefore, effective in dealing with behaviors that function as requests. Aggressive behavior negatively impacts children with ASD and their caregivers, decreasing their quality of life, increasing stress levels, and reducing educational and social support [34]. Thus, these findings highlight the importance of a favorable facility environment for residents and those who support them. 

Stereotyped behaviors also decreased in frequency after movement. Comparan-Meza et al. (2021) suggested that one of the factors that cause stereotypic behavior was an individual’s impairment in behavioral inhibition, cognitive flexibility, and monitoring [35]. As previously stated, the new facility was designed to minimize unnecessary stimuli and create a predictable environment. Therefore, residents were less likely to overreact to unneeded or unpredictable stimuli, as these features help address impairments in behavioral inhibition, cognitive flexibility, and monitoring. This may have contributed to the reduction in stereotyped behaviors. In addition, stereotyped behavior has also been linked to anxiety [36]. In an environment with daily task variability, anxiety about uncertainty or unwanted demands may be heightened. Furthermore, exposure to loud and unpredictable sounds can increase anxiety. The new facility’s format is believed to offer residents a more predictable environment by providing private rooms, clearly defining each room’s purpose within the unit, and minimizing stimulation from other residents. It was suggested that these characteristics are thought to help alleviate stereotypical behaviors associated with anxiety.

However, this study has some limitations. First, the results were obtained from a single facility with a limited sample size. The reliability and validity of the results could be enhanced by including users from other facilities that underwent similar renovations. Moreover, recruiting a larger number of subjects would facilitate the examination of differences in changes based on the type of behavioral disorder. Second, a control group was not established, making it challenging to rule out the possibility that changes in behavior occurred naturally over time, rather than solely due to environmental changes. Future studies should involve a larger number of participants from multiple facilities and quantitatively and qualitatively assess long-term changes in behavior. A larger participant pool may also aid in identifying the types and functions of challenging behavior that are particularly responsive to environmental adjustments. Given that residents spend extended periods in these facilities, tracking longer-term behavioral changes is essential.

## 5. Conclusions

This study found that residents’ CB was reduced one and six months after transitioning to a unit-care residential facility, with five residents living together in one unit and two private rooms per resident. This study is one of the few examining the effects of changes and considerations in the physical environment on behavioral disorders. More data from other facilities with similar renovations and different participants are further required to demonstrate the usefulness of environmental considerations for CB.

## Figures and Tables

**Figure 1 healthcare-12-00288-f001:**
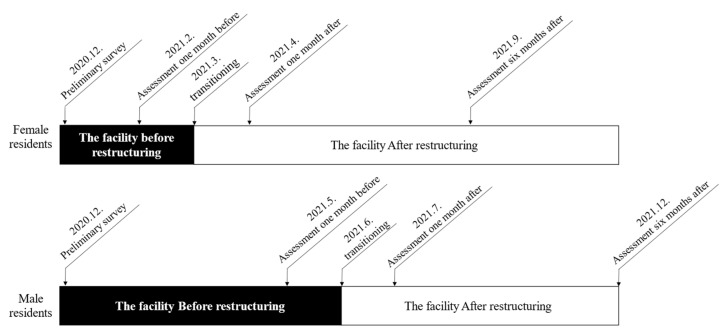
Study timeline.

**Figure 2 healthcare-12-00288-f002:**
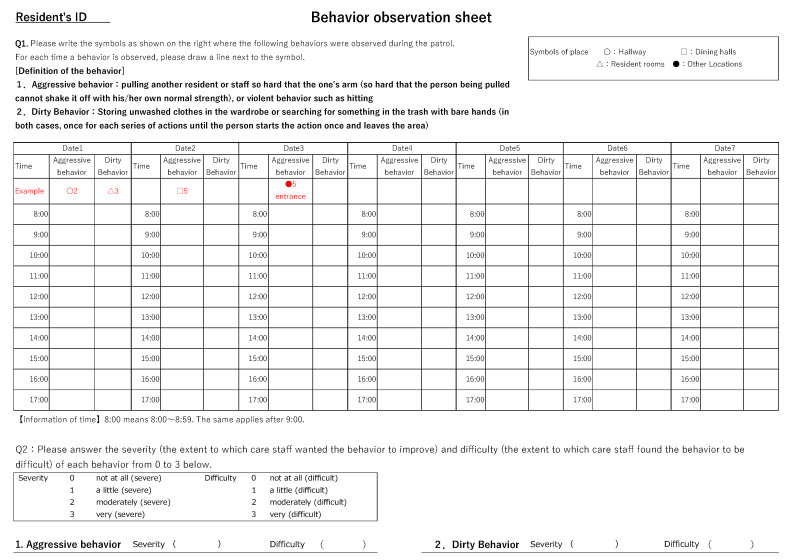
Behavior observation sheets.

**Figure 3 healthcare-12-00288-f003:**
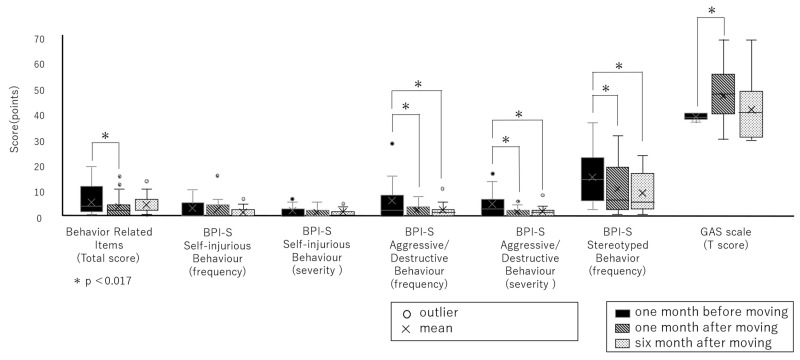
Changes in scores for each scale.

**Table 1 healthcare-12-00288-t001:** Facility features before and after restructuring.

		Before Restructuring	After Restructuring
Resident apartments	Total number	2 private apartments, 22 apartments for use by 2–3 people	44 private apartments
	Floor space per apartment	11.9 m^2^	14.2 m^2^
	Number of partitioned rooms per resident apartment	1	2 (bedroom and living room)
	Entrance	No signs	Signs featuring pictures the resident’s likes
	Number of residents per apartment	1–3	1
Restrooms	Total number	6 restrooms (men’s ward 4, women’s ward 2)	20 (including 4 changing rooms)
	Floor space	13.3 m^2^	5 m^2^
	Number of toilets per apartment	3	1
	Toilet type	Urinals and Japanese-style toilets	Urinals and Western-style toilets
Dining halls	Number of people dining together	10–15	5
	Floor space	78.1 m^2^	26 m^2^

**Table 2 healthcare-12-00288-t002:** Basic characteristics of participants.

Variable	Categories	n (%) or M ± SD
Sex	Male	17 (81.0)
	Female	4 (19.0)
Age (year)		35.7 ± 12.1
Length of stay (month)		132.3 ± 147.9
ASD	presence	11 (52.4)
	absence	10 (47.6)
Intellectual Disabilities ^1^	profound	15 (71.4)
	severe	4 (19.0)
	moderate	1 (4.8)
	no answer	1 (4.8)
Epilepsy	presence	2 (9.5)
	absence	19 (90.5)
Down syndrome	presence	1 (4.8)
	absence	20 (95.2)

^1^ Level of intellectual disability is based on the answers toi the Rehabilitation Certificate.

## Data Availability

All data generated or analyzed during this study are included in the published article.

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
