# Peer review of "Changes in Challenging Behaviors Accompanying Transition to a New Facility in Individuals with Intellectual Disabilities"

_healthcare, 2024, doi:10.3390/healthcare12030288_

Round 1
Reviewer 1 Report
Comments and Suggestions for Authors
PAST tense is not used consistently in some places
Line 13 - a difficult part of the behavioral aspect of severe
Needs rephrased by a native English speaker because it is not an aspect, it is the behaviour of interest
Line 21 - targeted behaviors.
These were not specified
Line 23 - These new features have eliminated unnecessary stimuli and clarified
Needs rephrased by a native English speaker
Line 36 - Children with CB are more likely to experience abuse [5].
This sentence is not integrated well with the previous sentence.
Line 39 - and factors of personal and physical environments
Needs rephrased by a native English speaker; problem term is ‘factors’
Line 40 - Therefore, environmental accommodations tailored to disability characteristics effectively reduce CB [6].
Needs rephrased by a native English speaker
Line 61 – considering the history of residential facility formats,
Needs rephrased by a native English speaker
Line 77- live
Change to….. lived
Line 111 –
The area were
Change to….. The area was
Line 118-
Each unit has
Change to…. Each unit had
Line 129-130
were subjected to behavioral observations.
Change to…. were observed.
Line 138
Certificate of Intellectual Disability needs a reference
Line 149
Behavior-Related Items needs a reference
Line 153
Aberrant Behavior Checklist (ABC-J) needs a reference
Line 159
Problems Inventory-Short Form (BPI-S) needs a reference
Figure 1
Timeline issues: The gap in the figure indicting 1 month after transition is similar for both Females and Males, but this is not the case for the gap between one month and 6 months.
Line 179
Authors need to provide this definition for the readers
Line 185-186
Show example of the recording sheet
Line 195-199
These terms are not defined in text so there is no way to understand their relevance to the calculation. Define them and explain with a reference the source of this equation.
Line 204
Therefore, the significance level was adjusted to p<0.017
This needs explained better.
Table 2
This is difficult to read because of spacing and because the text in centered.
Line 252-253
Need elaboration to understand the terms used
Line 262
It is not clear how this conclusion was made. Elaborate.
Line 273
It is not clear what ‘posting’ means.
Line 279-283
This sentence is not integrated with the previous sentence and therefor there is no way to understand the point being made.
Line 287-288
These accommodations address the factors of cognitive flexibility and anxiety linked to stereotypical behaviors.
This needs expanded upon. How is this conclusion derived?
Line 289-291
This last paragraph is not well integrated with the general text and reads like an after thought rather than a serious point being made.
Comments on the Quality of English Language
In general, the English is OK. However, there are a few places where I have requested that phrases be written by a native English speaker.
Author Response
Reply to the comments of the Reviewer No. 1
Thank you very much for evaluating our manuscript and for dedicating your valuable time to its review. We are sincerely grateful to reviewer 1 for their critical comments and helpful suggestions, which have significantly enhanced the quality of our paper. In the revised version (highlighted in red), we have diligently incorporated all of these comments and suggestions. We hope that the revised manuscript meets your approval.
Comment: PAST tense is not used consistently in some places
Response: Thank you for your valuable comment. I have revised the entire text and subsequently had it reviewed by a native speaker.
Line 13-14
Comment: “a difficult part of the behavioral aspect of severe” Needs rephrased by a native English speaker because it is not an aspect, it is the behaviour of interest.
Response: Thank you for addressing this issue. We have rephrased “challenging behavior” to describe it as a commonly observed behavioral problem, rather than emphasizing its difficulty within the behavioral aspect.
Line 21-22
Comment: “targeted behaviors”. These were not specified
Response: Thank you for pointing this out. We have provided examples of targeted behaviors, “such as hitting their own head and fecal smearing.”
Line 23-24
Comment: “These new features have eliminated unnecessary stimuli and clarified.” Needs rephrased by a native English speaker
Response: Your valuable comment is greatly appreciated. We have revised the sentence as follows: “These new features enable residents to reduce destructive stimuli and make it easier to understand what to do in each private room”.
Line 36-37
Comment: “Children with CB are more likely to experience abuse [5].” This sentence is not integrated well with the previous sentence.
Response: Thank you for bringing this to our attention. We have made the following changes to better integrate the text: “Because CB makes it difficult for caregivers to raise children, it can be a risk factor for abuse.”
Line 39
Comment: “and factors of personal and physical environments” Needs rephrased by a native English speaker; problem term is ‘factors’
Response: Thank you for addressing this issue. Following the advice of a native English speaker, we have revised the sentence as follows: “and physical and social environmental factors.”
Line 40-41
Comment: “Therefore, environmental accommodations tailored to disability characteristics effectively reduce CB [6].” Needs rephrased by a native English speaker.
Response: Thank you for raising this point. Following the advice of a native English speaker, we have added the word “specific” to the sentence.
Line 76
Comment: “considering the history of residential facility formats,” Needs rephrased by a native English speaker
Response: I appreciate your attention to this matter. To establish a stronger connection between the sentences, were have made the following change: “search for studies on residential facility formats.”
Line 94
Comment: “live” Change to….. lived
Response: Thank you for your important suggestion. We have made the changes as you suggested.
Line 124
Comment: “The area were” Change to….. The area was
Response: Thank you for your insightful suggestion. We have made the recommended changes. In addition, we have reevaluated and replaced the term "apartment" rather than "area" as it is more appropriate to describe a private room where each resident lives.
Line 130
Comment: “Each unit has” Change to…. Each unit had
Response: Your meaningful comment is noted. We have made the changes as you suggested.
Line 142-143
Comment: “were subjected to behavioral observations.” Change to…. were observed.
Response: We value your perceptive comment. The original sentence was redundant and did not effectively convey the content. Therefore, we have revised the sentence structure to "observed."
Line 155, 425-426
Comment: Certificate of Intellectual Disability needs a reference
Response: We are thankful for your valuable contribution. We have added a reference [20] for the “Certificate of Intellectual Disability.”
Line 166, 427-428
Comment: Behavior-Related Items needs a reference
Response: Your insightful comment is recognized and appreciated. We have added a reference [21] for the “Behavior-Related Items.”
Line 168, 429-430
Comment: Aberrant Behavior Checklist (ABC-J) needs a reference
Response: Your constructive input is greatly appreciated. We have added a reference [22] for the “Aberrant Behavior Checklist (ABC-J).”
Line 173,434-436
Comment: Problems Inventory-Short Form (BPI-S) needs a reference
Response: We extend our gratitude for your valuable feedback. We have incorporated reference [24] on the development of BPI-S into the relevant sections.
Line 178 Figure 1
Comment: Timeline issues: The gap in the figure indicting 1 month after transition is similar for both Females and Males, but this is not the case for the gap between one month and 6 months.
Response: Thank you for your valuable comment. We have equalized the gap between 1 month and 6 months after the transition for both females and males.
Line 195-201
Comment: Authors need to provide this definition for the readers
Response: We are grateful for your valuable remark. Following your suggestion, we have added an example to define the behavior of one resident, making it easier for readers to understand.
Line 208,226
Comment: Show example of the recording sheet
Response: We appreciate your insightful input. An example of the recording sheet was inserted in the image.
Line 208-210, 219-224
Comment: These terms are not defined in text so there is no way to understand their relevance to the calculation. Define them and explain with a reference the source of this equation.
Response: Thank you for providing these insights. The terms used to convert GAS scores to T-scores were explained in the immediately previous paragraph. Therefore, we presented reference 29 at the location of the formula and added a paragraph about the analysis (Line 220) and thereason of the use of T-scores (Lines 208-210). In addition, we have revised the explanation of the formula (Lines 219-225).
Line 233-236
Comment: “Therefore, the significance level was adjusted to p<0.017” This needs explained better.
Response: Thank you for your comment. We have included additional details explaining the reasons for adjusting the significance level.
Line 257 Table 2
Comment: This is difficult to read because of spacing and because the text in centered.
Response: Your efforts in resolving this matter are acknowledged. We have shifted the text to the left and adjusted the size of the space.
Line 286-291, 294-298, 299-305
Comment: Need elaboration to understand the terms used
Response: We are grateful for your efforts in rectifying this concern. Following you suggestion, we have added explanations for the terms, and we have restructured the entire sentence to discuss the results for each term.
Line 305-307
Comment: “As the new facility was designed with due consideration of these aspects, it could have helped decrease problematic behaviors one month after moving into the new facility.” It is not clear how this conclusion was made. Elaborate.
Response: Your prompt response to this issue is noted. We have revised the explanation from Lines 286 to 305 to clarify the basis for drawing this conclusion.
Line 318
Comment: “because of the consistent staff posting” It is not clear what ‘posting’ means.
Response: We thank you for addressing this issue promptly. The word “staff” has been changed to “staffing,” referring to the allocation of staff members to specific locations.
Line 324-331
Comment: This sentence is not integrated with the previous sentence and therefore there is no way to understand the point being made.
Response: We would like to express our thanks for your attention to this issue. We have revised the sentence to connect the results of previous studies with those of this study.
Line 332-338
Comment: “These accommodations address the factors of cognitive flexibility and anxiety linked to stereotypical behaviors.” This needs expanded upon. How is this conclusion derived?
Response: We are thankful for your efforts in resolving this matter. We have added an explanation for the basis of drawing this conclusion.
Line 339-350
Comment: “As this was a single-facility survey, the results were limited by the small sample size and lack of a control group. Future studies should increase the number of participants and examine long-term changes in behavior quantitatively and qualitatively.” This last paragraph is not well integrated with the general text and reads like an after thought rather than a serious point being made.
Response: Your dedication to addressing this matter is acknowledged. We have presented the study limitations clearly and provided explanations for future prospects corresponding to each limitation.
Comments on the Quality of English Language
In general, the English is OK. However, there are a few places where I have requested that phrases be written by a native English speaker.
Response: Thank you for your insightful comment. I had a native speaker review the revised full text.

Reviewer 2 Report
Comments and Suggestions for Authors
The abstract provides a comprehensive overview of the study, outlining the transition from the conventional facility to the redesigned format and its impact on the challenging behaviors exhibited by individuals with intellectual disabilities. It effectively communicates the objectives, methodology, and results of the research.
The introduction provides a comprehensive background and context for the study on challenging behavior (CB) among individuals with severe intellectual disabilities or autism spectrum disorder (ASD). It effectively highlights the prevalence of CB, its impact on health, caregivers' emotional reactions, and the relationship between disability characteristics, environmental factors, and CB manifestation.
The text articulates various studies that support the impact of environmental accommodations on reducing CB, citing examples of interventions tailored to learning styles associated with ASD and improvements seen with sensory processing accommodations. It also discusses the influence of living environments on resident behaviors, emphasizing the benefits of a person-centered approach, physical environment, and unit-based care on reducing CB severity and enhancing residents' quality of life.
However, the introduction could be improved in a few areas. Firstly, while it provides an extensive review of relevant literature, it could benefit from a more concise and focused discussion. Streamlining the information to prioritize key studies and findings that directly relate to the research's novelty and objectives would enhance its clarity and relevance.
Secondly, the introductory section lacks a clear statement of the research gap or the specific aim of the current study until the latter part of the text. Incorporating a more explicit statement about the unique contribution of the study and its aims earlier in the introduction would better orient the reader.
Lastly, the introduction's structure appears slightly disjointed, transitioning abruptly between discussions on environmental accommodations, historical facility formats, and the gap in existing research. Organizing these aspects in a more cohesive manner and transitioning between topics more smoothly would enhance the overall flow and readability of the text.
In conclusion, while the introduction provides a comprehensive background, it would benefit from greater focus, a clearer articulation of the research gap and objectives, and improved structural coherence to better guide the reader through the study's context and significance.
The "Materials and Methods" section provides a detailed description of the study facility, participant information, measures, and statistical analysis procedures employed in the research. The detailed account allows for transparency and replicability of the study's methodology.
Strengths:
· Description of Facilities: The detailed comparison between the facility before and after restructuring provides a clear understanding of the environmental changes made, aiding comprehension of their potential impact on resident behaviors.
· Participant Selection Criteria: The explanation of the criteria used to select residents, their consent process, and behavioral problems observed provides clarity regarding the study's sample.
· Measures and Tools: The description of tools used for data collection, such as the Behavior-Related Items scale, Behavior Problems Inventory-Short Form (BPI-S), and behavior observation sheets, demonstrates the thoroughness of the assessment methods employed.
Areas for Improvement:
· Conciseness: The detailed description of the facility's layout, especially before restructuring, might be overly exhaustive. Condensing non-essential details while maintaining key information could improve the section's readability without compromising its comprehensiveness.
· Explanation of Statistical Analyses: While the statistical analysis methodology is briefly described, providing a bit more detail about the rationale behind choosing specific statistical tests or how they relate to addressing the research questions would enhance the understanding of the analytical approach.
· Ethical Approval: While the study was approved by the Ethics Committee, it would be beneficial to include a brief statement regarding informed consent obtained from the participants or their legal guardians.
The "Results" section presents valuable findings regarding the basic characteristics of participants and changes observed in behavior-related scales following the restructuring of the facility.
Strengths:
· Clarity of Findings: The section provides clear and concise information about the basic characteristics of the selected residents, their behavioral profiles, and the changes observed in behavior-related scales.
· Numerical Representation: The presentation of quantitative data in tables and the use of statistical measures (e.g., mean, standard deviation, percentages) enhance the clarity and comprehension of the results.
The "Discussion" section provides a comprehensive overview of the study's findings and connects them with existing literature.
Strengths:
· Interpretation of Results: The section effectively interprets the observed behavioral changes in the context of facility restructuring and draws connections with relevant research. It offers plausible explanations for the observed behavioral modifications.
· Literature Integration: The references to previous studies effectively support the discussion by providing insights into similar findings and theories, enhancing the credibility of the interpretations.
· Highlighting Implications: The discussion appropriately emphasizes the potential significance of the facility environment in mitigating challenging behaviors, thereby indicating the practical implications for residents and caregivers.
Areas for Improvement:
· Enhanced Detail: While the section successfully links the facility environment to behavioral changes, providing more specific details about how certain environmental modifications directly impacted behavioral alterations would further enrich the discussion.
· Elaboration on Limitations: Although the study limitations are briefly mentioned, a more comprehensive discussion on the limitations, such as the constraints of a single-facility survey and the absence of a control group, would offer a clearer perspective on the study's scope and potential biases.
· Further Suggestions for Future Research: While the conclusion briefly mentions the need for additional data from similar facility renovations, providing more specific recommendations for future research directions, such as focusing on different types of facilities or diverse participant demographics, would strengthen the conclusion's depth.
Author Response
Thank you very much for evaluating our manuscript and for dedicating your valuable time to it. We have made efforts to incorporate your suggestions, which are highlighted in red within the revised manuscript. We hope that these revisions meet with your approval.
Comment 1: The introduction could be improved in a few areas. Firstly, while it provides an extensive review of relevant literature, it could benefit from a more concise and focused discussion. Streamlining the information to prioritize key studies and findings that directly relate to the research's novelty and objectives would enhance its clarity and relevance.
Secondly, the introductory section lacks a clear statement of the research gap or the specific aim of the current study until the latter part of the text. Incorporating a more explicit statement about the unique contribution of the study and its aims earlier in the introduction would better orient the reader.
Lastly, the introduction's structure appears slightly disjointed, transitioning abruptly between discussions on environmental accommodations, historical facility formats, and the gap in existing research. Organizing these aspects in a more cohesive manner and transitioning between topics more smoothly would enhance the overall flow and readability of the text.
In conclusion, while the introduction provides a comprehensive background, it would benefit from greater focus, a clearer articulation of the research gap and objectives, and improved structural coherence to better guide the reader through the study's context and significance.
Response: Thank you for your helpful and detailed advice regarding the Introduction. Your insights have significantly improved the clarity of the study's purpose and enhanced the overall coherence of the text. In the previous manuscript, the gap with previous studies was only addressed in the last paragraph. However, I have restructured the text to discuss this gap within the paragraphs on behavioral disorders and visual support, as well as behavioral disorders and sensory stimulation. Additionally, we have refined the discussion, making it more concise and focused. Moreover, we have introduced conjunctions and sentences to enhance the connections between paragraphs (Lines 41-43, 45-48, 51-54, 63-69).
Comment 2: Conciseness: The detailed description of the facility's layout, especially before restructuring, might be overly exhaustive. Condensing non-essential details while maintaining key information could improve the section's readability without compromising its comprehensiveness.
Response: We wish to convey our appreciation for your response to this issue. To maintain a concise restructuring, we have removed sentences with limited relevance to the results and discussion (Lines 102-117).
Comment 3: Explanation of Statistical Analyses: While the statistical analysis methodology is briefly described, providing a bit more detail about the rationale behind choosing specific statistical tests or how they relate to addressing the research questions would enhance the understanding of the analytical approach.
Response: Your timely response to this issue is commendable. We have added more detail on the reasons for conducting the Wilcoxon signed rank test and adjusting the significance level. Additionally, we have clarified the purpose of the study in the last paragraph of the Introduction (Lines 96-98, 227-236).
Comment 4: Ethical Approval: While the study was approved by the Ethics Committee, it would be beneficial to include a brief statement regarding informed consent obtained from the participants or their legal guardians.
Response: Thank you for pointing this out. As per your suggestion, we have included in section 2.2 (Participants) that we obtained consent from the facility director and legal guardians (Lines 139-143).
Comment 5: Enhanced Detail: While the section successfully links the facility environment to behavioral changes, providing more specific details about how certain environmental modifications directly impacted behavioral alterations would further enrich the discussion.
Response: We agree with your advice. We have completely rewritten the discussion section, providing detailed explanations of the previous study's findings and their connection to the results of this study (Lines 286-305, 324-338).
Comment 6: Elaboration on Limitations: Although the study limitations are briefly mentioned, a more comprehensive discussion on the limitations, such as the constraints of a single-facility survey and the absence of a control group, would offer a clearer perspective on the study's scope and potential biases.
Response: Thank you for providing these insights. We have added the study limitations you suggested (Lines 339-350).
Comment 7: Further Suggestions for Future Research: While the conclusion briefly mentions the need for additional data from similar facility renovations, providing more specific recommendations for future research directions, such as focusing on different types of facilities or diverse participant demographics, would strengthen the conclusion's depth.
Response: Your valuable comment is greatly appreciated. It has prompted us to consider the future development of this research. We have added future perspectives based on the limitations of the study (Lines 346-350).
